# Thermodynamic principle to enhance enzymatic activity using the substrate affinity

Hideshi Ooka [1] ✉, Yoko Chiba [1,2] & Ryuhei Nakamura [1,3]

Understanding how to tune enzymatic activity is important not only for biotechnological applications, but also to elucidate the basic principles guiding the design and optimization of biological systems in nature. So far, the Michaelis-Menten equation has provided a fundamental framework of enzymatic activity. However, there is still no concrete guideline on how the parameters should be optimized towards higher activity. Here, we demonstrate that tuning the Michaelis-Menten constant ($K_m$) to the substrate concentration ([S]) enhances enzymatic activity. This guideline ($K_m = $[S]) was obtained mathematically by assuming that thermodynamically favorable reactions have higher rate constants, and that the total driving force is fixed. Due to the generality of these thermodynamic considerations, we propose $K_m = $[S] as a general concept to enhance enzymatic activity. Our bioinformatic analysis reveals that the $K_m$ and in vivo substrate concentrations are consistent across a dataset of approximately 1000 enzymes, suggesting that even natural selection follows the principle $K_m = $[S].

Enzymes are responsible for catalysis in virtually all biological systems[1,2], and a rational framework to improve their activity is critical to promote biotechnological applications. Since the early 20th century, a reaction mechanism where the enzyme first binds to the substrate (E + S → ES) before releasing the product (ES → E + P) has been used as the conceptual basis to understand enzyme catalysis (Fig. 1)[3–6]. The reaction rate of this mechanism is given by the Michaelis-Menten equation:

$$v = \frac{k_2[S]}{K_m + [S]}\left[E_T\right].\tag{1}$$

Here, the reaction rate ($v$) is expressed as a function of a rate constant ($k_2$), the Michaelis-Menten constant ($K_m$), and the concentrations of the substrate ([S]) and enzyme ([$E_T$]). $K_m$ can be interpreted as a quasi-equilibrium constant for the formation of the enzyme-substrate complex, defined as:

$$K_m \equiv \frac{k_{1r} + k_2}{k_1},\tag{2}$$

with rate constants defined based on the mechanism shown in Fig. 1. $k_2$ is the rate constant for releasing the product from the enzyme-substrate complex (ES → E + P), routinely expressed as $k_{cat}$ in the enzymology literature. These parameters are experimentally accessible by fitting the theoretical rate law (Eq. (1)) with experimental data[7–10] and are subsequently registered in databases such as BRENDA[11] and Sabio-RK[12]. The accumulated data may help rationalize and improve the activity of existing enzymes.

However, rational improvement of enzymatic activity is difficult, because a quantitative understanding on how the kinetic parameters influence enzymatic activity is missing. For example, increasing $k_2$ will enhance activity according to Eq. (1) if no other parameters are changed. However, changing $k_2$ will increase $K_m$ according to Eq. (2),

[1]Biofunctional Catalyst Research Team, Center for Sustainable Resource Science, 2-1 Hirosawa, Wako, Saitama 351-0198, Japan. [2]Faculty of Life and Environmental Science, University of Tsukuba, 1-1-1 Tennoudai, Tsukuba, Ibaraki 305-8577, Japan. [3]Earth-Life Science Institute (ELSI), Tokyo Institute of Technology, 2-12-IE-1 Ookayama, Meguro-ku, Tokyo 152-8550, Japan. ✉e-mail: hideshi.ooka@riken.jp

**Fig. 1 | Mechanism of a standard enzymatic reaction.** The enzyme (E) and substrate (S) form a complex (ES) which then releases the product (P). Symbols above the arrows indicate rate constants.

**Fig. 2 | The free energy landscape corresponding to the mechanism shown in Fig. 1.** The free energy changes ($\Delta G_1$, $\Delta G_2$) and activation barriers ($E_{a1}$, $E_{a1r}$, $E_{a2}$) in the mathematical analysis are defined as indicated in the figure.

which is unfavorable for activity[13]. Furthermore, if $k_2$ is increased by making the second step (ES → E + P) more thermodynamically favorable, this would come at the expense of the first step (E + S → ES) because the free energy available for the entire reaction (S → P) is fixed. In such a case, $k_1$ would decrease, which is unfavorable for activity. Thus, the mutual dependence between $k_2$, $K_m$, and other kinetic parameters complicates their influence on the enzymatic activity $v$. Understanding how to optimize these parameters under thermodynamic restrictions would clarify the physical limits achievable in enzyme catalysis, and would lead to the rational design of enzymes towards biotechnological applications such as the synthesis of commodity chemicals[14], antibiotics[15], or pharmaceuticals[16], increasing the nutritional content of crops[17], and restoring the environment[18].

In this study, we analyzed the Michaelis-Menten equation under basic thermodynamic constraints to clarify the relationship between the enzyme-substrate affinity ($K_m$) and the activity ($v$). The main consideration is that the free energy difference between the substrate and the product ($\Delta G_T$) is fixed, while the enzyme is allowed to optimize the free energy difference between the substrate and the enzyme-substrate complex ($\Delta G_1$). To bridge thermodynamics with kinetic parameters such as $k_2$ or $K_m$, we have used the Brønsted (Bell)-Evans-Polanyi (BEP) relationship[19–23], which models the activation barrier as a function of the driving force. This is a well-known concept in heterogeneous catalysis, and in conjunction with the Arrhenius equation[24], can be used to evaluate the mutual dependence between $k_2$ and $K_m$ to quantitatively. This allowed us to calculate the optimum value of $K_m$ required to maximize enzymatic activity ($v$), a finding which is supported by our bioinformatic analysis of approximately 1000 wild-type enzymes.

## Results

### Construction of the thermodynamic model

In principle, an ideal enzyme with low $K_m$ and large $k_2$ can be realized if both $k_1$ and $k_2$ are increased simultaneously. However, this is physically unrealistic, because the driving force which can be allocated to $k_1$ and $k_2$ is limited by the free energy change of the entire reaction. Within this thermodynamic context, maximum activity is realized by optimizing the distribution of the total driving force between the first (E + S → ES) and second (ES → P) steps shown in Fig. 1. To quantitatively evaluate the relationship between the driving force and the activity, we have used the BEP relationship[19–23] to convert driving forces ($\Delta G$) into activation barriers ($E_a$), and the Arrhenius[24] equation to convert activation barriers to rate constants.

The thermodynamic model which served as the basis of our calculations is shown in Fig. 2. In a classical Michaelis-Menten reaction, the enzyme and substrate first form an enzyme-substrate complex (E + S → ES) before producing the product in the second step (ES → E + P). This mechanism is conceptually similar to reactions that occur on a heterogeneous catalyst surface, where the substrate molecule first binds to the catalyst surface before being converted into the product[19–23]. The Gibbs free energies for the formation of the enzyme-substrate complex and the product are denoted as $\Delta G_1$ and $\Delta G_2$, respectively. By definition, their sum must equal the total free energy change of the reaction $\Delta G_T$:

$$\Delta G_T = \Delta G_1 + \Delta G_2. \tag{3}$$

To evaluate the reaction rate under this thermodynamic restriction, a method to convert thermodynamics ($\Delta G_1$, $\Delta G_T$) to kinetics and rate constants is necessary. One possibility is to use the BEP relationship, which is a well-known empirical rule in heterogeneous catalysis[19–23]. This relationship suggests that a thermodynamically unfavorable elementary reaction will have a larger activation barrier[19–23]. For example, the activation barrier corresponding to $k_1$ can be expressed mathematically as:

$$E_{a1} = E_{a1}^0 + \alpha_1 \Delta G_1, \tag{4}$$

where $E_{a1}^0$ represent the activation barriers when the elementary reaction is in equilibrium ($\Delta G_1 = 0$), and $\alpha_1$ expresses how sensitive the activation barrier is with respect to the driving force. The applicability of the BEP relationship to enzymes is supported by the bioinformatic analysis by Sousa et al.[25], who found a linear relationship between activation barriers and driving forces of 339 wild type hydrolases. Similar linear relationships have also been reported experimentally for cellulases[26] and computationally for cytochrome P-450[27], suggesting that the BEP relationship may be applicable to a wide variety of enzymes.

Next, activation barriers can be converted to rate constants based on the Arrhenius equation[24] as follows:

$$k_1 = A_1 \exp \frac{-E_{a1}}{RT}. \tag{5}$$

Here, $A_1$ is a pre-exponential factor, and $R$ and $T$ are the gas constant and absolute temperature, respectively. Using Eqs. (4) and (5), $k_1$ can be expressed as:

$$\begin{aligned} k_1 &= k_1^0 \exp \frac{-\alpha_1 \Delta G_1}{RT} \\ &= k_1^0 g_1^{-\alpha_1}, \end{aligned} \tag{6}$$

where $k_1^0 \equiv A_1 \exp \frac{-E_{a1}^0}{RT}$ and $g_1 \equiv \exp \frac{\Delta G_1}{RT}$ were used to aggregate factors independent and dependent on the driving force, respectively (see Supplementary Note 1, Appendix 1 for details). $k_{1r}$ and $k_2$ can also be

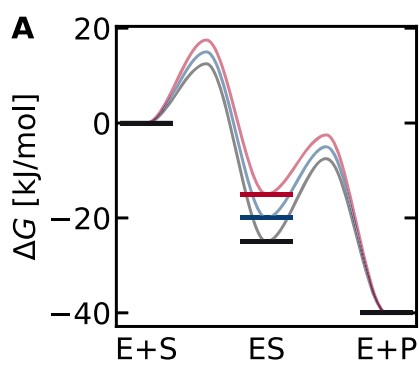

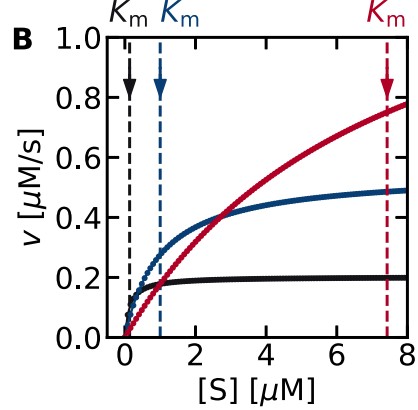

**Fig. 3 | Relationship between thermodynamic landscapes and enzymatic activity.** Three thermodynamic landscapes are shown in **A**. Their corresponding Michaelis-Menten plots are shown in **B**. The $K_m$ values are indicated as vertical dashed lines in **B**. Increasing the driving force of the first step increases the activity at low substrate concentrations but lowers the activity at high substrate concentrations. Therefore, the thermodynamic landscape of an optimum enzyme depends on the substrate concentration ([S]). The free energies of the enzyme-substrate complex ($\Delta G_1$) were −25, −20, and −15 kJ/mol for the black, blue, and red lines, respectively, and that of the total reaction ($\Delta G_T$) was −40 kJ/mol. All numerical simulations in this study were performed at $[E_T] = 0.01\,\mu M$, $k_1^0 = k_2^0 = 1$ (1/μM/s and 1/s units, respectively), and $\alpha_1 = \alpha_2 = 0.5$ unless otherwise noted. See the python code in Supplementary Data 2 for details.

written similarly as:

$$k_{1r} = k_1^0 g_{1r}^{\alpha_{1r}} = k_1^0 g_1^{1-\alpha_1}, \tag{7}$$

$$k_2 = k_2^0 g_2^{-\alpha_2} = k_2^0 \left(\frac{g_1}{g_T}\right)^{\alpha_2}. \tag{8}$$

using notations similar to those defined for $k_1$ (See Appendices 2 and 3 for details). Substituting these rate constants into Eq. (2) yields the following expression for $K_m$:

$$K_m \equiv \frac{k_{1r} + k_2}{k_1} \tag{9}$$
$$= g_1(1+K),$$

where $K$ was defined as $K \equiv \frac{k_2^0 g_1^{\alpha_1+\alpha_2-1}}{k_1^0 g_T^{\alpha_2}}$. Finally, based on Eqs. (8) and (9), the enzymatic activity ($v$) can be expressed as:

$$v = \frac{k_2[S]}{K_m + [S]}[E_T] \tag{10}$$
$$= \frac{k_2^0 g_1^{\alpha_2} g_T^{-\alpha_2}[S]}{g_1(1+K) + [S]}[E_T].$$

To illustrate how Eq. (10) captures the tradeoff relationship between $k_2$ and $K_m$, numerical simulations were performed (Fig. 3A). Three possible thermodynamic landscapes for a reaction with a total driving force of $\Delta G_T = -40$ kJ/mol are shown. This parameter was chosen as a representative value based on the fact that the $\Delta G_T$ of typical biochemical reactions is between −80 and +40 kJ/mol[28,29]. Similar calculations with different values of $\Delta G_T$ can be found in Supplementary Notes 2 (Supplementary Figs. 1–3). When the first reaction is thermodynamically favorable compared to the second ($\Delta G_1 < \Delta G_2$; Fig. 3A, black lines), the activity increases rapidly from low substrate concentrations (Fig. 3B, solid black line), consistent with the small $K_m$ value. However, an enzyme with a small $K_m$ suffers from a small $k_2$ value, which is evident from the saturating behavior at [S] >1 μM. Increasing the driving force of the second step (blue and red lines) leads to a larger $k_2$ and thus higher activity at large [S] (>1 μM) compared to the enzyme shown in black. However, in this case, the activity at low [S] (>1 μM) is suppressed due to the larger $K_m$.

The influence of the substrate concentration in Fig. 3. can be rationalized by considering the rate-limiting step. At low substrate concentrations, the rate of the first step (E + S→ ES: $k_1[E][S]$) would be diminished due to the small [S]. This suggests that spending more driving force on the first step (Fig. 3, black line) such that it is no longer rate-limiting would be favorable for overall activity. On the other hand, at high substrate concentrations (Fig. 3, red line), the first step is already kinetically favored, and it becomes more beneficial to spend more driving force on the second step. The boundary condition is when the rates of the two forward reactions are equal ($k_1[E] \cdot [S] = k_2[ES] \leftrightarrow [S] = \frac{k_2[ES]}{k_1[E]}$. This boundary is valid as long as [S] can be assumed to be constant. In a batch reactor system, this would require $[E_T]$ to be small relative to [S][30]. However, in a flow reactor or under in-vivo conditions, the boundary holds for larger values of $[E_T]$ as long as the external supply of the substrate is sufficient to maintain [S] constant. Under these conditions, the optimum values of $k_1$ and $k_2$ are dependent on the substrate concentration, and thus, the $K_m$ value necessary to maximize the activity must also be dependent on [S].

### Analysis of the activity–driving force relationship
To directly illustrate the influence of driving force ($\Delta G_1$ and $\Delta G_T$) on enzymatic activity, we performed numerical simulations using Eq. (10) at various fixed substrate concentrations (Fig. 4). At a substrate concentration of 0.1 μM (Fig. 4a), the region of highest enzymatic activity (orange) was observed in the bottom left region. It is reasonable for activity to be higher in the lower half of the panel, due to the more negative $\Delta G_T$. A negative $\Delta G_1$ is also beneficial for activity at a low substrate concentration ([S] = 0.1 μM), leading to enzymatic activity being higher in the left half of the panel. At higher substrate concentrations, the overall color within each panel changed from blue to red, because a higher substrate concentration increases activity (Fig. 4b–d). At the same time, the $\Delta G_1$ corresponding to maximum activity gradually shifted positively (black dashed lines). This finding is consistent with Fig. 3 which shows that a more positive $\Delta G_1$ is desirable when the substrate concentration is increased. In all panels, the location with the highest activity at a given $\Delta G_T$ value is shown as a dashed black line. Notably, when the $K_m$ value was calculated at the ($\Delta G_1, \Delta G_T$) values under the dashed line using Eq. (9), the obtained value was always equal to the substrate concentration [S] in each panel. In other words, the dashed line is not only the ridge of the volcano plot, but is also the contour line showing $K_m = [S]$. This suggests that the condition for maximizing enzymatic activity can be represented by $K_m = [S]$. The

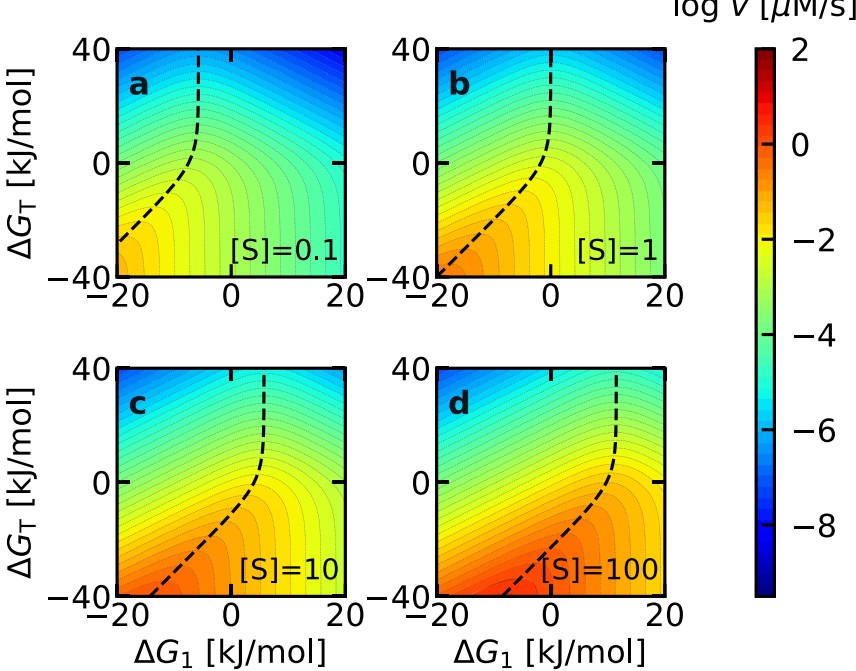

**Fig. 4 | Enzymatic activity ($v$) plotted against $\Delta G_1$ and $\Delta G_T$ based on Eq. (10).** The substrate concentration ([S]) in each panel was **a** $10^{-1}$, **b** 1, **c** 10, and **d** $10^2$ $\mu$M, as indicated in the bottom right of each panel. In all panels, the black dashed line corresponding to $K_m =$ [S] overlaps with the region of highest enzyme activity.

fact that $K_m =$ [S] yields high activity is valid even if the BEP coefficients deviate from 0.5 (Supplementary Note 4, see also below).

To examine why $K_m =$ [S] leads to maximum activity, Eq. (10) was rearranged to give the following expression for the activity ($v$):

$$v = \frac{k_2^0 g_T^{-\alpha_2}[S]}{[S]g_1^{-\alpha_2} + g_1^{1-\alpha_2} + \frac{k_2^0 g_1^{\alpha_1}}{k_1^0 g_T^{\alpha_2}}}[E_T], \qquad (11)$$

where $g_1$ is only in the denominator. The derivative of the denominator, denoted as $f$ is:

$$\frac{df}{dg_1} = -\alpha_2 g_1^{-(\alpha_2+1)}[S] + (1-\alpha_2)g_1^{-\alpha_2} + \frac{k_2^0 \alpha_1}{k_1^0 g_T^{\alpha_2}}g_1^{\alpha_1-1}. \qquad (12)$$

To maximize the activity ($v$), $f$ must be minimized which is realized at:

$$\frac{df}{dg_1} = 0 \leftrightarrow [S] = g_1\left(\frac{1-\alpha_2}{\alpha_2} + \frac{\alpha_1}{\alpha_2}K\right). \qquad (13)$$

Using standard notation, the condition for the optimum thermodynamic landscape is given by:

$$\Delta G_1 = RT\left(\ln[S] - \ln\left(\frac{1-\alpha_2}{\alpha_2} + \frac{\alpha_1}{\alpha_2}K\right)\right). \qquad (14)$$

In the specific case of $\alpha_1 = \alpha_2 = 0.5$, Eq. (14) reduces to:

$$\Delta G_1 = RT(\ln[S] - \ln(1+K)). \qquad (15)$$

The condition $\alpha_1 = \alpha_2 = 0.5$ corresponds to a scenario where the activation barriers in the forward and backward directions change equally with respect to the driving force. In general, if the BEP coefficient is large ($\alpha > 0.5$), the forward direction is more sensitive, while if $\alpha < 0.5$, the backward reaction is more sensitive. For reversible enzymes[31,32], large deviations from $\alpha = 0.5$ would hinder their ability to

catalyze the reaction in both directions. Furthermore, typical experimental values of $\alpha$ range between 0.3 and 0.7 for artificial catalysts[33–35], and the experimental value reported for cellulases is 0.74[26]. Therefore, we expect the unbiased scenario ($\alpha = 0.5$) to be a reasonable representation for the median value of enzymes in general. Setting BEP coefficients to 0.5 is also a common technique used to understand general trends in heterogeneous catalysis[22,36–38].

Under this condition, substituting $K$ in Eq. (13) using the definition of $K_m$ ($K_m \equiv g_1(1+K)$, Eq. (10)), yields a surprisingly simple formula for the condition of maximum activity:

$$K_m = [S] \qquad (16)$$

This equation shows that the combination of ($\Delta G_1, \Delta G_T$) necessary to maximize the activity guarantees $K_m = $ [S]. This finding is further illustrated in Fig. 5, where the activity ($v$) is plotted as a function of $K_m$ at different substrate concentrations. In all cases, maximum activity ($v$) is observed when the binding affinity ($K_m$) is equal to the substrate concentration ([S]). Kari et al. have reported that the activity of cellulases[26,39] and PET hydrolases[40], are maximized at a specific $K_m$. However, the physical origin of this trend was unclear, due to the difficulty in obtaining raw $\Delta G$ values from experiments. As $K_m$ is a composite parameter which depends on multiple rate constants, only relative values of the free energy ($\Delta\Delta G$) have been discussed so far. In this study, we have started from the thermodynamic landscape and have shown that as long as the enzyme kinetics can be expressed using the Michaelis-Menten equation (Eq. (1)), and the rate constants follow the BEP relationship with $\alpha_1 = \alpha_2 = 0.5$, tuning the $K_m$ value equal to the substrate concentration [S] guarantees maximal enzymatic activity. The existence of an optimum $K_m$ close to the substrate concentration is valid even under mechanistic deviations as will be shown below.

**Robustness of the theoretical model**

To confirm the robustness of our finding, we have performed numerical simulations by loosening each of the theoretical requirements. Deviation from the Michaelis-Menten mechanism (Fig. 1) are

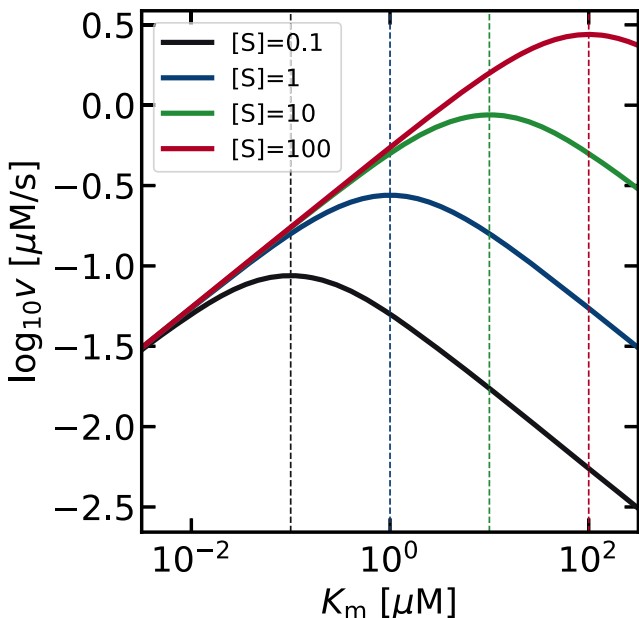

**Fig. 5 | Volcano plots showing how the activity is expected to change with respect to the Michaelis-Menten constant ($K_m$).** As the substrate concentration was increased from $10^{-1}$ $\mu$M (black) to $10^2$ $\mu$M (red), the volcano plot shifted to the upper right. The apex is located at $K_m = $[S], as indicated by the vertical dashed lines of the corresponding color. Changing the values of $\Delta G_T$, $k_1^0$ or $k_2^0$ does not influence the conclusion that the activity is maximized when $K_m = $[S], as shown in Supplementary Note 3.

shown in Fig. 6a–c, and deviation of $\alpha$ values from 0.5 are shown in Fig. 6d. The possibility of reverse reactions (P → S) or inhibition (E + I → EI or ES + I → ESI) are common deviations from Michaelis-Menten kinetics[41]. The net rate in the presence of a reverse reaction when the substrate and product are in equal concentrations ([S] = [P] = 10 $\mu$M) is shown in Fig. 6a. In terms of maximizing the activity in the forward direction (S → P), the physically meaningful region is ($\Delta G_T < 0$), where the net reaction proceeds in the forward direction. Under this condition, the dashed line corresponding to $K_m = $[S] and the solid line corresponding to the true maximum activity (forward minus reverse reaction rates) overlap almost completely, indicating that $K_m = $[S] is a good guideline to enhance activity even in the presence of reverse reactions (P → S).

Similar calculations for competitive and uncompetitive inhibition, where the inhibitor binds to either the free enzyme or the enzyme-substrate complex, are shown in Fig. 6b, c. The degree of inhibition ($\gamma \equiv \frac{[I]}{K_i}$), is determined by the inhibitor concentration ([I]) and the equilibrium constant of inhibition ($K_i$)[41]. Based on the experimental data of Park et al.[42], $\gamma$ can range from $10^{-4}$ to $10^4$. As $\gamma$ was less than 10 in approximately 80% of their data, $\gamma = 10$ was used here for the numerical simulations. Again, the optimal $K_m$ (solid line) deviates only slightly from the dashed line ($K_m = $[S]), and both lines pass through the region of high activity (orange). The $K_m$ values are approximately 1 order of magnitude apart between dashed and solid lines, yet there is only a 57 % difference in activity at a specific $\Delta G_T$. This is much smaller than the scale of the entire diagram (10 orders of magnitude), suggesting that adjusting $K_m$ to the substrate concentration [S] is a robust strategy to enhance the activity, even in the presence of inhibition. A detailed discussion on the parameter dependence ($\gamma$, [S]), as well as for other mechanisms such as substrate inhibition or allostericity

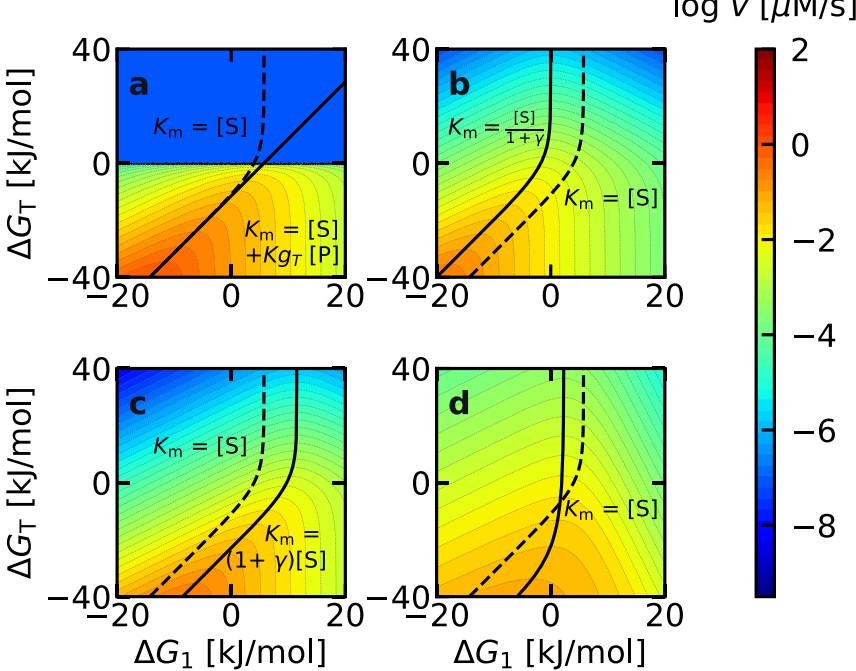

**Fig. 6 | Influence of mechanistic deviations on the optimum $K_m$. a** Reverse reactions, **b** Competitive inhibition, **c** Uncompetitive inhibition, and **d** BEP coefficient ($\alpha$). The dashed line corresponds to $K_m = $[S], with [S] = 10 $\mu$M. The true optimum $K_m$ for each mechanism is shown as a solid line along with its analytical equation (refer to Supplementary Note 5 for the derivations). In **a**, the product concentration ([P]) was set to 10 $\mu$M. The top half of **a** was colored at an arbitrarily

low activity because the reverse reaction is more favorable in this region. The large discrepancy between the dashed and solid lines at $\Delta G_T > 0$ is physically irrelevant, because the activity of the forward reaction cannot be discussed when the net reaction proceeds in the reverse direction. In **b** and **c**, the degree of inhibition ($\gamma \equiv I/K_i$) was set to 10. In **d**, the BEP coefficients were set to $\alpha_1 = \alpha_2 = 0.2$. No analytical solution was obtained for **d**.

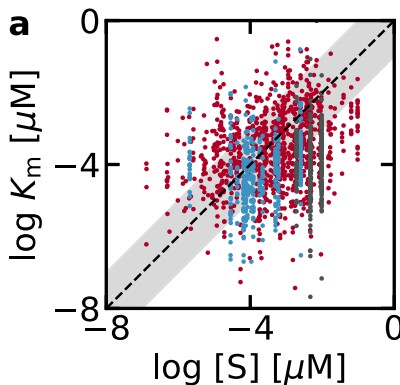

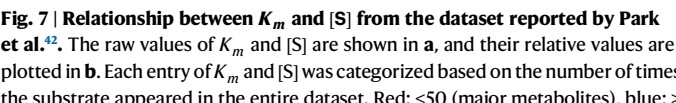

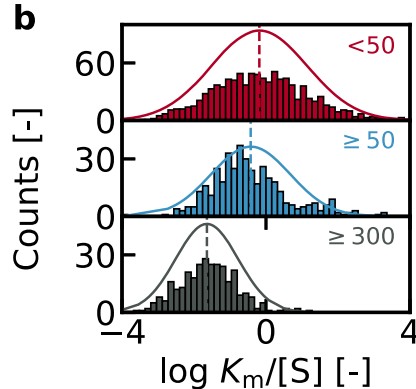

**Fig. 7 | Relationship between $K_m$ and [S] from the dataset reported by Park et al.[42].** The raw values of $K_m$ and [S] are shown in **a**, and their relative values are plotted in **b**. Each entry of $K_m$ and [S] was categorized based on the number of times the substrate appeared in the entire dataset. Red: <50 (major metabolites), blue: > 50 (NAD⁺, NADH, NADP⁺, NADPH, and acetyl-CoA), black: > 300 (ATP). The number of entries was used as a proxy for the validity of the Michaelis-Menten mechanism of the specific substrate. The dashed line in **a** corresponds to $K_m$ = [S], and the shaded area shows a deviation of 1 order of magnitude.

can be found in Section 5 of the supporting information. The derivations for the equations of the true optimal $K_m$ can also be found in the same section.

The influence of the assumption $\alpha_1 = \alpha_{1r} = \alpha_2 = 0.5$ is shown in Fig. 6d. As physical constraints require $\alpha_{1r} = 1 - \alpha_1$ (Appendix 2), only $\alpha_1$ and $\alpha_2$ are independent. In an extreme case of $\alpha_1 = \alpha_2 = 0.2$, the activity is diminished because rate constants hardly change even if their driving force is increased. However, the dashed line still passes through the region of high activity, and the activity is still less than an order of magnitude away from the true optimum (solid line). The fact that $K_m$ = [S] yields high activity is valid even for other values of $\alpha_1$ and $\alpha_2$ (Supplementary Figure 5). This dashed line was obtained through numerical optimization, because no analytical solution for the optimum $K_m$ was obtained for general values of $\alpha_1$ and $\alpha_2$. We note that the optimality obtained by Kari et al.[26] ($\frac{\alpha_2}{1-\alpha_2}$[S]) is a special case of Eq. (13), which can be obtained under the assumption $\alpha_1 + \alpha_2 = 1$. Further assuming $\alpha_1 = \alpha_2 = 0.5$ yields $K_m$ = [S]. Taken together, the simulations confirm that $K_m$ = [S] is a robust theoretical guideline to enhance enzymatic activity.

### Validation based on experimental data

Finally, to evaluate whether $K_m$ = [S] can rationalize enzymatic properties in nature, we have analyzed their relationship based on the experimental data from Park et al[42]. The original data (Supplementary Data 1) consisted of $K_m$ values of wild-type enzymes obtained from BRENDA, and intracellular [S] values obtained from *Escherichia coli*, *Mus musculus*, and *Saccharomyces cerevisiae* cells, yielding a total of 1703 $K_m$–[S] combinations. This dataset was then classified by the number of entries for each substrate, based on the expectation that a substrate which participates in many reactions is more likely to deviate from Michaelis-Menten kinetics under in-vivo conditions. For example, the Michaelis-Menten mechanism does not consider scenarios where multiple enzymes compete for the same substrate, a situation which may occur for cofactors such as ATP. Major metabolites, such as sugars or amino acids all appear less than 50 times each in the dataset and are shown in red. The comparison between their raw $K_m$ and [S] values (Fig. 7a), as well as the histogram of their relative values (Fig. 7b) indicate that the distribution is centered around $K_m$ = [S]. Namely, the $K_m$ and [S] are consistent to within 1 order of magnitude for 53% of this dataset (524 out of 980 entries), and the Gaussian distribution fitted to the histogram is centered at $\log_{10} K_m/[S] = -0.18$ with a standard deviation of 1.3.

The large standard deviation is due to a variety of factors, such as inhibitors or BEP coefficients which can change the optimum $K_m$ by roughly an order of magnitude (Fig. 6), or growth conditions and measurement errors which may influence [S] also by an order of

magnitude[43]. Furthermore, some enzymes are outside the applicability domain of our model. For example, some enzymes do not follow the BEP relationship at all[25], and in some cases, the Michaelis-Menten equation may be an inadequate expression of enzymes under in-vivo conditions. Namely, the Michaelis-Menten equation is derived traditionally based on the assumption that the concentration of the enzyme-substrate complex is in the steady state, but this assumption can be broken if the amount or activity of the enzyme is so high such that the substrate is quickly depleted[30]. Superoxide dismutase from bovine blood is one example, as its high activity ($k_{cat} = 1.9 \times 10^9$ M⁻¹ s⁻¹) renders it to be diffusion-limited[44] under physiological conditions. Accordingly, it deviates from our proposed law: $K_m$ = [S] with a $K_m$ (> 0.5 mM)[44] several orders of magnitude larger than the substrate concentration (25 <[H₂O₂]< 60 μM in aqueous humor)[45]. Not all superoxide dismutases are exceptions, as those with lower activity ($k_{cat} < 3 \times 10^8$ M⁻¹ s⁻¹) from *Thermus thermophilus* ($K_m$ = 30.8 μM)[46] and *Escherichia coli* ($K_m$ = 75 μM)[47] have $K_m$ values closer to the substrate concentration. Previous studies[48] have shown that diffusion limited enzymes are not the majority, suggesting that our proposed law may apply to the majority of enzymes. Within our dataset, only 1% (10 entries) show $K_m/[S] > 10^3$.

The next subset shown in blue contains 410 entries and consists of 5 substrates which each appear more than 50 times: NAD⁺, NADH, NADP⁺, NADPH, and acetyl-CoA. The Gaussian fitted to the histogram is slightly shifted to smaller $K_m$ (centered at $\log_{10} K_m/[S] = -0.43$), but 57% of this dataset (232 out of 410 entries) still satisfies $K_m$ = [S] to within an order of magnitude. On the other hand, ATP, which is the most frequently occurring substrate with 313 entries, shows a significant deviation from $K_m$ = [S]. The fitted Gaussian is centered at $\log_{10} K_m/[S] = -1.64$, and $K_m$ is smaller than [S] for 98% of the entries. The deviation from $K_m$ = [S] may be because the Michaelis-Menten mechanism, which is the basis of our mathematical analysis, does not consider scenarios where multiple enzymes compete for the same substrate. Under such conditions, the effective substrate concentration available to each enzyme would decrease. Thus, $K_m \ll$ [ATP] may be a result of $K_m$ being adjusted to the effective substrate concentration. Although activity is not the only enzymatic property that must be optimized in nature, the consistency between the $K_m$ of wild-type enzymes and in-vivo substrate concentrations suggests that natural selection does indeed favor enzymes which satisfy $K_m$ = [S], the theoretical guideline for achieving high enzymatic activity.

### Discussion

So far, various criteria[13,41,49] such as large $k_2$ ($k_{cat}$), small $K_m$, or large $k_2/K_m$ have been proposed to characterize enzymes with high activity,

making it difficult to rationally evaluate or improve the activity of an enzyme. The lack of a universal consensus is largely due to the mutual dependence between $k_2$ and $K_m$. As our theoretical model addresses this challenge directly and maximizes the activity within the thermodynamic constraints imposed by $k_2$ and $K_m$, we believe that $K_m = [S]$ is a criterion for high activity which provides the optimum balance between $k_2$ and $K_m$ in a wider range of scenarios.

As to the limitations of our theory, we note that the mathematical equations derived in this study are based on the empirical BEP relationship, and therefore, $K_m = [S]$ may not yield maximum enzymatic activity in scenarios where the BEP relationship is broken. Possible strategies include tuning the local binding environment using 3 dimensional active sites[50–52], or by using the Marcus inverted region in redox reactions[53,54]. Furthermore, the starting point of our analysis is the Michaelis-Menten equation. Traditionally, this equation has been derived based on the steady state approximation of the enzyme-substrate complex[30]. Therefore, if this assumption is broken such as in the case of diffusion-limited enzymes[44], $K_m$ and $[S]$ may diverge by several orders of magnitude. Recently, several studies have explicitly addressed the differential equations of Michaelis-Menten and similar enzyme mechanisms to determine the exact applicability domain of the Michaelis-Menten equation[30,55,56]. For example, Schnell[30] has proposed that instead of the steady-state approximation of [ES], the reactant stationary assumption is the true condition for the Michaelis-Menten equation to be applicable. In this case, the applicability domain of our theory would also adhere to that of the Michaelis-Menten equation. Other deviations in the mechanism (Fig. 6a–c) or parameter values (Fig. 6d) do not significantly influence the activity landscape[30].

Our main conclusion that the Michaelis-Menten constant should be increased at higher substrate concentrations to maximize activity is consistent with the experimental work by Kari et al.[39], who measured the activity of cellulases with different $K_m$. When the substrate concentration was increased 6 times, the $K_m$ value of the most active enzyme increased approximately 2.4 times. Considering that their $K_m$ had a range of roughly 3 orders of magnitude, the experimental trend supports our hypothesis $K_m = [S]$, especially when their experimental BEP coefficient of 0.74 is also considered. The idea of the optimum binding affinity being dependent on the reaction condition and driving force is also consistent with recent theoretical models of heterogeneous catalysis[22,57–59].

As a corollary, our model which quantifies the relationship between $K_m$ and $k_2$ immediately provides a thermodynamic rationale to the recently reported scaling relationship between them in cellulases[26]. Namely, for general values of $\alpha_1$ and $\alpha_2$, the relationship between $K_m$ and $k_2$ can be written as:

$$K_m = (1+K)g_T \left(\frac{k_2}{k_2^0}\right)^{1/\alpha_2} \tag{17}$$

$$\therefore \log k_2 = \alpha_2 \log K_m - \alpha_2 \log(1+K)g_T + \log k_2^0.$$

This equation shows that $\log k_2$ and $\log K_m$ are linearly correlated by a factor of $\alpha_2$, and provides a physical basis not only to the high linearity ($R^2 = 0.95$) observed for cellulases[26], but also to the reason behind why it is generally difficult to realize enzymes with high $k_2$ ($k_{cat}$) and small $K_m$. Even highly active enzymes operating near the diffusion limit seem to have difficulty in breaking such scaling relationships, because although their $k_2$ is extremely large ($k_{cat} > 10^6$ s$^{-1}$), their $K_m$ is also generally large ($K_m > 1$ mM), and as a result, $k_{cat}/K_m$ cannot exceed $10^9$ s$^{-1}$M$^{-1}$ [50]. The consistency between our theoretical model and previously accumulated experimental insight suggests that it may be possible to quantitatively rationalize enzymatic properties based on fundamental principles of physical chemistry.

## Methods

The mathematical formulas were derived by hand, and the step-by-step derivations for the standard Michaelis-Menten mechanism are explained in the main text. The derivations in the presence of inhibition and allostericity are provided in the Supplementary Information. Numerical simulations and bioinformatic analysis were performed using Python 3.8.3.

## Data availability

The bioinformatics data obtained from the supporting information of ref. 42. is available as Supplementary Data 1. It can also be accessed at https:github.com/HideshiOoka/SI_for_Publications and has been deposited to Zenodo[60].

## Code availability

The main code used for the numerical simulations can be obtained as Supplementary Data 2. All code used in this study, such as for performing the bioinformatic analysis or generating the figures can be found at https:github.com/HideshiOoka/SI_for_Publications and has been deposited to Zenodo[60].

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

## Acknowledgements

H.O. gratefully acknowledges the support from the JST FOREST program (Grant Number JPMJFR213E, Japan). Y.C. is grateful for the support from the JST ACT-X program (Grant Number JPMJAX20BB, Japan).

## Author contributions

H.O. performed the mathematical calculations, numerical simulations, and bioinformatic analysis. H.O., Y.C., and R.N. conceived this work and wrote the manuscript.

## Competing interests

The authors declare no competing interests.
