## [Peer Review File · Nature Communications]

REVIEWER COMMENTS

Reviewer #1 (Remarks to the Author):

General description of the work.

Hideshi Ooka, Yoko Chiba, and Ryuhei Nakamura introduce a novel model to rationalize steady-state enzyme kinetics that combines conventional Michaelis-Menten kinetics with concepts from heterogeneous catalysis. By applying the Brønsted (Bell)-Evans-Polanyi scaling relation to the MM-equation they show by simulations that their model can explain why many enzymes have evolved to have a K_m close to the substrate concentration ($K_m \sim S_0$). An empirical observation that has not yet been sufficiently accounted for by existing theories on enzyme catalysis. The authors show how their theory can explain this phenomenon for moderately efficient enzymes. The work is both relevant and novel and the authors also demonstrate the robustness of their theory by expanding it to both reversible reactions and enzyme reaction with inhibition. Their main punchline is that the optimal enzyme has a K_m which is (approximately) equal to the substrate concentration. This simple but powerful statement may have significant implications for both enzyme design and discovery. The paper is very original and the subject is of broad interest to the community. However, I have some concerns regarding the authors main point, $K_m \sim S_0$, and the generality of this statement (my main concern).

General recommendations

Overall, the work is excellent and should be published in Nature Communication if the following issues can be addressed.

1) Model assumptions

1.1 BEP relationship (BEP assumption)

The novelty of the work is its application of the BEP relations in the MM-equations. However, it is also the main critique since little experimental evidence exists that enzymes are governed by BEP relationships. The justification of the model comes from its ability to predict why many enzymes have K_m values close to the physiological substrate concentration. However, if the proposed model is intended as a general model alongside the classical MM-model then the authors should present more experimental evidence (from literature) that can back up their claim. It is difficult to imagine that enzymes that work under diffusion limitations should be governed by BEP relationships since such enzymes usually have a very high K_m , much higher than their physiological substrate

concentrations (1). Further, it has been argued that 3D active sites with a local binding environment such as that of an enzyme may be a way to break linear scaling relationships (such as BEP) for heterogeneous catalysts(2). The authors should be more precise about which type of enzymes their model may be applied to.

1.2 MM-model (steady-state assumption)

The MM-equation is only valid if the steady-state assumption holds. It can be shown the steady-state assumption is valid is $E_0/(S_0 + K_m) \ll 1$ (3). In the contour plot, dashed contour lines indicate conditions where $E_0/(S_0 + K_m) = E_0/2S_0$ (since $K_m=S_0$). Hence, the MM-equation is valid along the contour line is $E_0/2S_0 \ll 1$. Setting a threshold for 'much smaller than unity' to 0.1 implies that $E_0/2S_0 \leq 0.1$ or $E_0 \leq 0.2S_0$. Thus the enzyme concentration should be approximately 1/5 of the substrate concentration at the contour line (assuming that $K_m \sim S_0$). For many of the simulations, this requirement is violated since the enzyme concentration in the simulations seems to be 1 μM and the substrate concentration in e.g. Figure 3 is 0.1 μM , 1 μM , 10 μM and 100 μM . The authors should emphasize this point in the manuscript as the model is only valid for experimental conditions where $E_0 \leq 0.2S_0$. If experimentalists are going to design assays to screen for better enzymes using the design principle proposed by the authors, then such information is important. Further, the authors should rerun the simulations using a lower enzyme concentration such that the requirement $E_0/(S_0 + K_m) \ll 1$ is fulfilled in all the simulations.

Minor points

Figure 2

The authors should add energy barriers to the energy diagram to the left. All three enzymes would behave identically without barriers as the intermediate is redundant for consecutive "downhill" reactions without "kinetic-traps". Since $a_1=r_1=r_2=0.5$, all three barriers (E_{a1} , E_{a1} and E_{a2}) are known from the BEP relationship and could easily be added to the energy diagram. I acknowledge that the authors have chosen this representation to focus on the thermodynamic driving forces and their effects (right panel), but it is confusing since the effect comes from the coupling between the driving force and the barriers (the BEP relationship).

Page 6

Line 3: add "fixed" to the sentence "...performed numerical simulations using Eq. (10) at various fixed substrate concentrations.."

Line 9: remove "always" "...because a higher substrate concentration always increases activity.."

Symbolic notations

To avoid confusion the authors should use e.g. square brackets [] when they refer to the concentration of the different species in the MM-model (S, E, ES, P, I). It is confusing when the authors both use the letters as a symbol for the species and as species concentrations. E.g. $K_m = S$ should be written $K_m = [S]$.

References

1. Fersht, A. (1999) Structure and Mechanism in Protein Science: A Guide to Enzyme Catalysis and Protein Folding, W. H. Freeman
2. Vojvodic, A., and Nørskov, J. K. (2015) New design paradigm for heterogeneous catalysts. National Science Review 2, 140-143
3. Schnell, S. (2014) Validity of the Michaelis-Menten equation-steady-state or reactant stationary assumption: that is the question. Febs Journal 281, 464-472

Reviewer #2 (Remarks to the Author):

In this manuscript, the authors argue that an optimality principle ($K_m = S$) for enzymatic activity can arise from a phenomenological relationship between the activation energy and the energy difference of reactants and products, which may hold for catalyzed reactions within a common class.

This is an interesting topic, but I have two major concerns, one regarding the substance of the paper itself, and the second concerning the novelty / added value of this work.

First, the arguments leave me unconvinced about how fundamental this $K_m = S$ condition is. Beginning in the introduction, I feel the authors do not frame very cleanly the optimization problem that $K_m \sim S$ is supposed to be solving---in particular *under what constraints* on fundamental parameters is $K_m = S$ supposed to maximize activity? The early discussion of the mutual dependence of k_2 and K_m is confused because if you increase k_2 in the MM mechanism holding all other rate parameters constant, v always increases, even though K_m is decreasing in k_2 . Later on, it becomes clear that the key constraint they impose to derive $K_m = S$ is $a = 0.5$. But I want to emphasize that they do not provide a physical argument for this choice, saying only that it is a "common

assumption". Varying the value of a breaks the optimality relation $K_m = S$. The bioinformatic analysis they present, if I understand it correctly, is also not especially consistent with $K_m = S$ being a principle seen in nature.

Second, I am concerned about the contribution of this work relative to that of Kari et al. "Physical constraints and functional plasticity of cellulases" Nat Commun (2021). The present authors cite this work, mostly giving the impression that it is an experimental work, but I was surprised to find that Kari et al. not only provide the optimality principle (in the form $K_m = S a / (1-a)$), but also a detailed discussion and thermodynamic derivation of where it may come from---an explanation which is substantially the same as (and in my view, better presented) that of the present authors. Importantly, the arguments of Kari et al. seem to me possess at least the same level of generality and rigor as those of the present authors. The present authors assume $a = 0.5$, to get $K_m = S$.

The authors should make much clearer what exactly a reader, especially one familiar with Kari et al., ought to learn from this work. If it is, for example, their numerical simulations loosening the MM assumptions or their bioinformatic analysis, then e.g. this part of the work should be headlined.

Reviewer #1 (Remarks to the Author)

We have renumbered the comments to clarify our point-by-point response. Responses to this reviewer are highlighted in yellow and green. Notational changes (S to $[S]$, etc) were not highlighted due to their frequent occurrence. The notational change was also adopted in this reply. Minor English revisions in the manuscript were highlighted in grey. Punctuations were added to equations.

Overall Comment

General description of the work

Hideshi Ooka, Yoko Chiba, and Ryuhei Nakamura introduce a novel model to rationalize steady-state enzyme kinetics that combines conventional Michaelis-Menten kinetics with concepts from heterogeneous catalysis. By applying the Brønsted (Bell)-Evans-Polanyi scaling relation to the MM-equation, they show by simulations that their model can explain why many enzymes have evolved to have a K_m close to the substrate concentration ($K_m \approx S_0$), an empirical observation that has not yet been sufficiently accounted for by existing theories on enzyme catalysis. The authors show how their theory can explain this phenomenon for moderately efficient enzymes. The work is both relevant and novel and the authors also demonstrate the robustness of their theory by expanding it to both reversible reactions and enzyme reaction with inhibition. Their main punchline is that the optimal enzyme has a K_m which is (approximately) equal to the substrate concentration. This simple but powerful statement may have significant implications for both enzyme design and discovery. The paper is very original and the subject is of broad interest to the community. However, I have some concerns regarding the authors main point, $K_m \approx S_0$, and the generality of this statement (my main concern).

General recommendations

Overall, the work is excellent and should be published in Nature Communication if the following issues can be addressed.

Overall Reply

We appreciate the positive evaluation of our manuscript.

Comment 1-1) Model assumptions

Comment 1-1A BEP relationship (BEP assumption)

The novelty of the work is its application of the BEP relations in the MM-equations. However, it is also the main critique since little experimental evidence exists that enzymes are governed by BEP relationships. The justification of the model comes from its ability to predict why many enzymes have K_m values close to the physiological substrate concentration. However, if the proposed model is intended as a general model alongside the classical MM-model then the authors should present more experimental evidence (from literature) that can back up their claim.

Reply 1-1A

We appreciate this comment by the reviewer, as the BEP relationship is indeed a critical part of our model. In addition to the work by Kari et al. (*Nat. Commun.*, **2021**, *12*, 1.), there is a bioinformatic study (Sousa et al., *ACS Catal.* **2015**, *5*, 5877) which reported a linear correlation between activation barriers and driving forces of 126 wild type hydrolases. Additionally, cytochrome P-450 was also reported to exhibit a linear relationship between driving forces and activation barriers (Korzekwa et al., *JACS* **1990**, *112*, 7042). These studies suggest that the BEP may be applicable to a wide variety of enzymes. We have added the following sentence and citation to the main text to clarify this point (**page 4 lines 15-19, green highlights**):

Original

Recently, Kari et al have shown that fungal cellulases indeed satisfy such linear free energy relationships between the activation barrier and the driving force.

Revised

The applicability of the BEP relationship to enzymes is supported by the bioinformatic analysis by Sousa et al.,^[25] who found a linear relationship between activation barriers and driving forces of 339 wild type hydrolases. Similar linear relationships have also been reported experimentally for cellulases^[26] and computationally for cytochrome P-450,^[27] suggesting that the BEP relationship may be applicable to a wide variety of enzymes.

Furthermore, we have revised the following sentences to provide a more intuitive explanation of the BEP relationship (**page 4 lines 7-15, yellow highlights**):

Original

From these thermodynamic constraints, we will use the BEP relationship to obtain activation barriers (E_a), and then the Arrhenius equation to obtain rate constants, which ultimately yields quantitative insight on the relationship between k_1 , k_2 , and K_m . Based on the BEP relationship, the activation barrier corresponding to k_1 can be written as:

$$E_{a1} = E_{a1}^0 + \alpha_1 \Delta G_1, \quad (4)$$

Revised

To evaluate the reaction rate under this thermodynamic restriction, a method to convert thermodynamics (ΔG_1 , ΔG_T) to kinetics and rate constants is necessary. One possibility is to use the BEP relationship, which is a well-known empirical rule in heterogeneous catalysis.^[19-23] This relationship suggests that a thermodynamically unfavorable elementary reaction will have a larger activation barrier.^[19-23] For example, the activation barrier corresponding to k_1 can be expressed mathematically as:

$$E_{a1} = E_{a1}^0 + \alpha_1 \Delta G_1, \quad (4)$$

where E_{a1}^0 represent the activation barriers when the elementary reaction is in equilibrium ($\Delta G_1 = 0$), and α_1 expresses how sensitive the activation barrier is with respect to the driving force.

Comment 1-1B

It is difficult to imagine that enzymes that work under diffusion limitations should be governed by BEP relationships since such enzymes usually have a very high K_m , much higher than their physiological substrate concentrations (1).

Reply 1-1B

We appreciate the reviewer for giving us the chance to clarify the limitations of our theory. First, we believe that even enzymes working under diffusion limitation may show a correlation between the driving force and activation barriers (BEP as defined in Eq. (4)). Their k_{cat} and K_m are reported to be large (Bar-Even et al., *Biochem.* **2011**, *50*, 4402), which is consistent with Eq. (17) in the revised manuscript. As Eq. (17) was derived directly from Eq. (4) without any further assumptions such as the value of the BEP coefficient, we believe there is a possibility that even highly active enzymes may follow the BEP relationship.

On the other hand, we agree that their large K_m relative to the substrate concentration ($K_m \gg [S]$) is a direct contradiction to our proposed law: $K_m = [S]$. For example, superoxide dismutase has a $K_m > 0.5$ mM (Rotilio et al., *Biochim. Biophys. Acta*, **1972**, *268*, 605.), while the concentration of hydrogen peroxide in aqueous humor is between 25 and 60 μM (Bleau et al., *Anal. Biochem.* **1998**, *263*, 13). We believe that this is because for such enzymes, $[S]$ is depleted immediately, which breaks the “reactant stationary assumption” (Schnell, *FEBS J.* **2014**, *281*, 464) required for the Michaelis-Menten equation to be applicable. At least within the dataset of Bar-Even et al, which consists of approximately 2500 enzymes obtained from KEGG and BRENDA databases, diffusion limited enzymes ($k_{cat}/K_m > 10^8$ $\text{M}^{-1}\text{s}^{-1}$) are not the majority, as 60% of the enzymes operate far below the diffusion limit ($10^3 < k_{cat}/K_m < 10^6$ $\text{M}^{-1}\text{s}^{-1}$). Also within our dataset, only 1 % (10 entries out of 980) of the enzymes in our bioinformatic analysis which react with major metabolites show $[S]/K_m > 10^3$.

We have added the following sentence to clarify these points.

Sentence added (page 14, lines 32-35, green highlights)

Even highly active enzymes operating near the diffusion limit seem to have difficulty in breaking the BEP relationship, because although their k_2 is extremely large ($k_{cat} > 10^6 \text{ s}^{-1}$), their K_m is also generally large ($K_m > 1 \text{ mM}$), and as a result, k_{cat}/K_m cannot exceed $10^9 \text{ s}^{-1}\text{M}^{-1}$.

Sentence added (page 12, lines 20-26, green highlights)

Some enzymes operating near the diffusion limit are known to have K_m several orders of magnitude larger than $[S]$. For example, superoxide dismutase has a $K_m > 0.5 \text{ mM}$,^[44] while the concentration of hydrogen peroxide in aqueous humor is between 25 and 60 μM .^[45] This is a direct contradiction with our proposed law: $K_m = [S]$, which may originate from the fact that the Michaelis-Menten equation is not accurate for enzymes which quickly deplete the substrate (Schnell, FEBS J). However, previous studies^[30] have shown that they are not the majority, and even within our dataset, only 1 % (10 entries) in this dataset show $K_m/[S] > 10^3$.

Comment 1-1C

Further, it has been argued that 3D active sites with a local binding environment such as that of an enzyme may be a way to break linear scaling relationships (such as BEP) for heterogeneous catalysts (2). The authors should be more precise about which type of enzymes their model may be applied to.

Reply 1-1C

We agree with the reviewer that breaking the BEP relationship is possible. However, breaking the BEP is still considered an achievement in artificial catalysis (Mano et al., *JPCC*, **2023**, *127*, 7683, Darby et al., *JPCL*, **2018**, *9*, 5636, Gani et al., *ACS Catal.*, **2018**, *8*, 975), despite the BEP relationship being known for more than 80 years.

In order to break the BEP, there are two well-known strategies:

- a) 3D active sites which stabilize specific intermediates more than others, as the reviewer has proposed
- b) The Marcus inverted region, where a more negative ΔG increases E_a .

To clarify these points, we have added the following sentences in the discussion:

Paragraph added (page 14 lines 10-15, green highlights)

As to the limitations of our theory, we note that the mathematical equations derived in this study are based on the empirical BEP relationship, and therefore, $K_m = [S]$ may not yield maximum enzymatic activity in scenarios where the BEP relationship is broken. Possible strategies include tuning the local binding environment using 3 dimensional active sites, or by using the Marcus inverted region in redox reactions. Other deviations in the mechanism (Fig. 5A-C) or parameter values (Fig. 5D) do not seem to significantly influence the activity landscape.

1-2 MM-model (steady-state assumption)

Comment 1-2A

The MM-equation is only valid if the steady-state assumption holds. It can be shown the steady-state assumption is valid is $\frac{E_0}{S_0 + K_m} \ll 1$ (3). In the contour plot, dashed contour lines indicate conditions

where $\frac{E_0}{S_0 + K_m} = \frac{E_0}{2S_0}$ (since $K_m = S_0$). Hence, the MM-equation is valid along the contour line is $\frac{E_0}{2S_0} \ll$

1. Setting a threshold for ‘much smaller than unity’ to 0.1 implies that $\frac{E_0}{2S_0} \leq 0.1$ or $E_0 \leq 0.2 S_0$. Thus

the enzyme concentration should be approximately 1/5 of the substrate concentration at the contour line (assuming that $K_m \approx S_0$). For many of the simulations, this requirement is violated since the enzyme concentration in the simulations seems to be 1 μ M and the substrate concentration in e.g. Figure 3 is 0.1 μ M, 1 μ M, 10 μ M and 100 μ M. The authors should emphasize this point in the manuscript as the model is only valid for experimental conditions where $E_0 \leq 0.2 S_0$.

Reply 1-2A

We are grateful to the reviewer for the critical assessment of our theoretical model. We have decreased $[E]_0$ ($[E_T]$ in the main text) from 1 to 0.01 μ M, and now, all figures in the main text satisfy $[E]_0 < 0.2 [S]_0$.

However, we believe that our model is valid even at $[E]_0 \geq 0.2 [S]_0$ for the following reasons:

- a) Our main argument is that the reviewer (and the reference from Schnell, FEBS J. 2013) is considering the limitations in an experimental, batch-reactor setting where $[S]$ decreases with time. In this case, we agree that having a large $[E]_0$ such as $> 0.2 [S]_0$ would quickly deplete $[S]$ such that the steady-state approximation is not applicable. However, in our theoretical model, $[S]$ is kept constant with respect to time and is not depleted regardless of the value of $[E]_0$. Such a situation may be physically possible when there is an external supply of S, either in a flow reactor setting or under *in vivo* conditions where the reaction of $E+S \rightarrow ES \rightarrow E+P$ is only a small part of the entire metabolic network. In any case, if $[S]$ is constant, then $[ES]$ will reach a steady state concentration regardless of $[E]_0$. Therefore, while we have exchanged the figures in the main text, the figures are almost identical with the previous version. The only difference is the scale of the activity, which decreased by 2 orders of magnitude in response to the decrease of $[E_T]$.
- b) In the study cited by the reviewer (Schnell, FEBS J. 2013), the author himself mentions that:

“The reactant stationary assumption is a stronger condition than that required for the steady state assumption and is sufficient for the validity of the steady-state assumption.”

The reactant concentration is constant in our study, satisfying this requirement.

- c) Using the notation in the reference above, the steady state approximation is valid when $t_c \ll t_s$. Here, t_c and t_s are the timescales of $[ES]$ increase and $[S]$ decrease as defined by Schnell. In a batch reactor, t_s is finite. However, when $[S]$ is kept constant as a system parameter, t_s is infinite, and thus, the condition for steady-state approximation ($t_c \ll t_s$) is guaranteed.

To clarify the limitations of this approach for the reader, we have added the following sentences to the manuscript:

Sentence added (page 6, lines 7-10, yellow highlights)

This boundary is valid as long as $[S]$ can be assumed to be constant. In a batch reactor system, this would require $[E_T]$ to be small relative to $[S]$.^[30] However, in a flow reactor or under *in-vivo* conditions, the boundary holds for larger values of $[E_T]$ as long as the external supply of the substrate is sufficient to maintain $[S]$ constant.

Comment 1-2B

If experimentalists are going to design assays to screen for better enzymes using the design principle proposed by the authors, then such information is important.

Reply 1-2B

We agree that this information is important for the reader and have thus clarified the relevant experimental conditions in the manuscript (see Reply 1-2A).

Comment 1-2C

Further, the authors should rerun the simulations using a lower enzyme concentration such that the requirement $\frac{E_0}{S_0 + K_m} \ll 1$ is fulfilled in all the simulations.

Reply 1-2C

All figures in the main text which depend on the value of the enzyme concentration (Figs. 2-5) have been exchanged. All simulations were performed at 0.01 μM , which is an order of magnitude lower than the smallest enzyme concentration (0.1 μM).

Minor points

Comment 1-3 Figure 2

The authors should add energy barriers to the energy diagram to the left. All three enzymes would behave identically without barriers as the intermediate is redundant for consecutive “downhill” reactions without “kinetic-traps”. Since $\alpha_1 = \alpha_{1r} = \alpha_2 = 0.5$, all three barriers (E_{a1} , E_{a1r} , and E_{a2}) are known from the BEP relationship and could easily be added to the energy diagram. I acknowledge that the authors have chosen this representation to focus on the thermodynamic driving forces and their effects (right panel), but it is confusing since the effect comes from the coupling between the driving force and the barriers (the BEP relationship).

Reply 1-3

We have updated Figure 2 based on this comment. We are grateful to the comments by this reviewer, whose critical advice was helpful for conveying our message more clearly.

Comment 1-4

Page 6 Line 3: add “fixed” to the sentence “...performed numerical simulations using Eq. (10) at various fixed substrate concentrations..”

Page 6 Line 9: remove “always” “.because a higher substrate concentration always increases activity..”

Reply 1-4

Revised accordingly.

Comment 1-5 Symbolic notations

To avoid confusion the authors should use e.g. square brackets [] when they refer to the concentration of the different species in the MM-model (S, E, ES, P, I). It is confusing when the authors both use the letters as a symbol for the species and as species concentrations. E.g. $K_m = S$ should be written $K_m = [S]$.

Reply 1-5

We have added square brackets as recommended by the reviewer. In the original, we had distinguished species and their concentrations using upright and italic font, respectively (S and *S*), but we agree that the distinction is easier in the revised version with brackets.

This concludes our response to reviewer #1. Once again, we express our gratitude for the comments which have helped improve the clarity of our manuscript.

Hideshi Ooka (PhD)
hideshi.ooka@riken.jp

Reviewer #2 (Remarks to the Author)

We have numbered the comments to clarify our point-by-point response. Responses to this reviewer are highlighted in light blue and green. Minor English revisions are shown in grey. In the revised manuscript, we have changed notations such that concentrations are indicated in square brackets (S to $[S]$, etc). Punctuations were added to equations.

Overall Comment

In this manuscript, the authors argue than an optimality principle ($K_m = S$) for enzymatic activity can arise from a phenomenological relationship between the activation energy and the energy difference of reactants and products, which may hold for catalyzed reactions within a common class.

This is an interesting topic, but I have two major concerns, one regarding the substance of the paper itself, and the second concerning the novelty / added value of this work.

Overall Reply

We are grateful to the reviewer for the interest in our work, as well as the constructive comments which have helped us improve the clarity of our manuscript. Our point-by-point response to the substance (2-1) and novelty (2-2) can be found below.

Comment 2-1A

First, the arguments leave me unconvinced about how fundamental this $K_m = S$ condition is. Beginning in the introduction, I feel the authors do not frame very cleanly the optimization problem that $K_m \sim S$ is supposed to be solving---in particular *under what constraints* on fundamental parameters is $K_m = S$ supposed to maximize activity?

Reply 2-1A

This paper aims to clarify the ideal thermodynamic condition which will maximize enzymatic activity. The main constraint is the free energy difference between the substrate and product (ΔG_T). The free variable is the free energy difference between the substrate and the enzyme-substrate complex (ΔG_1). Eq. (13) gives the condition for maximum activity for general values of α . We then show that this thermodynamic condition is mathematically equivalent to $K_m = [S]$ when $\alpha_1 = \alpha_{1r} = \alpha_2 = 0.5$.

We have added the following sentence to clarify the constraint:

Sentence added (page 3 lines 3-5, blue highlights)

The main consideration is that the free energy difference between the substrate and the product (ΔG_T) is fixed, while the enzyme is allowed to optimize the free energy difference between the substrate and the enzyme-substrate complex (ΔG_1).

Comment 2-1B

The early discussion of the mutual dependence of k_2 and K_m is confused because if you increase k_2 in the MM mechanism holding all other rate parameters constant, v always increases, even though K_m is decreasing in k_2 .

Reply 2-1B

Once again, we appreciate the feedback which has helped us improve the clarity of our manuscript. Our argument is that it is generally difficult to “increase k_2 holding all other parameters constant”. The most straightforward way to increase k_2 is to apply a larger driving force (make ΔG_2 more negative). However, as the total free energy between the substrate and product is fixed, this reduces the free energy available for the first step, which may decrease k_1 . To circumvent the tradeoff between k_1 and k_2 , rate constants must be tuned independently of the driving force. In other words, the BEP relationship must be broken. This is possible in principle, but at least for 126 wild type hydrolases (Sousa et al., *ACS Catal.* **2015**, *5*, 5877), cellulases (Kari et al., *Nat. Commun.*, **2021**, *12*, 1.), and cytochrome P-450 (Korzekwa et al., *JACS* **1990**, *112*, 7042), their driving force and activation barriers are linearly correlated. We acknowledge that our theory cannot be applied in its current state to enzymes which break the BEP relationship.

We have revised the manuscript to clarify the difficulty of “increasing k_2 holding all other parameters constant” (**page 2, lines 29-35, blue highlights**):

Original

For example, increasing k_2 may enhance activity according to Eq. (1), or diminish it due to a larger K_m (Eq. (2)).

Revised

For example, increasing k_2 will enhance activity according to Eq. (1) if no other parameters are changed. However, changing k_2 will increase K_m according to Eq. (2), which is unfavorable for activity.^[13] Furthermore, if k_2 is increased by making the second step ($ES \rightarrow E+P$) more thermodynamically favorable, this would come at the expense of the first step ($E+S \rightarrow ES$) because the free energy available for the entire reaction ($S \rightarrow P$) is fixed. In such a case, k_1 would decrease, which is unfavorable for activity.

We have also added references supporting the applicability of the BEP relationship in enzymes as follows (**page 4 lines 15-19, green highlights**):

The applicability of the BEP relationship to enzymes is supported by the bioinformatic analysis by Sousa et al.,^[25] who found a linear relationship between activation barriers and driving forces of 339 wild type hydrolases. Similar linear relationships have also been reported experimentally for cellulases^[26] and computationally for cytochrome P-450,^[27] suggesting that the BEP relationship may be applicable to a wide variety of enzymes.

We have also explicitly mentioned that our theory cannot be applied to enzymes which deviate from the BEP relationship by adding a paragraph in the discussion section:

Paragraph added (page 14 line 10-15, green highlights)

As to the limitations of our theory, we note that the mathematical equations derived in this study are based on the empirical BEP relationship, and therefore, $K_m = [S]$ may not yield maximum enzymatic activity in scenarios where the BEP relationship is broken. Possible strategies include tuning the local binding environment using 3 dimensional active sites,^[47-49] or by using the Marcus inverted region in redox reactions.^[50,51] Other deviations in the mechanism (Fig. 5A-C) or parameter values (Fig. 5D) do not seem to significantly influence the activity landscape.

Comment 2-1C

Later on, it becomes clear that the key constraint they impose to derive $K_m = S$ is $\alpha = 0.5$. But I want to emphasize that they do not provide a physical argument for this choice, saying only that it is a “common assumption”.

Reply 2-1C

We agree with the reviewer that the physical reasoning behind $\alpha_1 = \alpha_2 = 0.5$ was lacking. Therefore, we have restructured the manuscript such that the main conclusions do not rely on this assumption.

First, we have added simulations when $\alpha \neq 0.5$ as Fig. S5. Even when $\alpha_1 = \alpha_2 = 0.3$ or 0.7 (Fig. S5A and D), the dashed line showing $K_m = [S]$ still passes through the high activity region (> 50% of maximum activity). A more extreme version ($\alpha_1 = \alpha_2 = 0.2$) of this calculation is also presented in the main manuscript (Fig. 5D). These results indicate that changing α from 0.5 does not significantly change the conclusion that $K_m = [S]$ is a reasonable strategy to enhance enzymatic activity.

Figure added as Fig. S5

Fig. S5. Enzymatic activity (v) plotted against ΔG_1 and ΔG_T based on Eq. (10). Parameters are the same as in Fig. 3 except for the BEP coefficients which was changed from $(\alpha_1, \alpha_2) = (0.3, 0.3)$, $(0.3, 0.7)$, $(0.7, 0.3)$, and $(0.7, 0.7)$ between panels A-D. The activity changes only 4 orders of magnitude in panel A, compared to 10 orders of magnitude in panel D due to the different sensitivity with respect to the driving force. However, the dashed line indicating $K_m = [S]$ still passes through the area of high activity ($> 50\%$ of maximum activity), indicating that it gives a reasonable direction to enhance enzymatic activity even when $\alpha_1 = \alpha_2 = 0.5$ is not satisfied.

We have also added the following sentence to guide the reader to this result:

Sentence added (page 11 line 21)

The fact that $K_m = [S]$ yields high activity is valid even if the BEP coefficients deviate from 0.5 (Fig. 5D, S5).

Second, we have realized from the reviewer's comment that the original manuscript sounded as if $\alpha_1 = \alpha_2 = 0.5$ is a prerequisite for our theory. In fact, it is only necessary for the simple expression "maximum activity at $K_m = [S]$ " to be correct. As mentioned above, "high activity at $K_m = [S]$ " is generally correct, even when $\alpha_1 = \alpha_2 = 0.2$ (Fig. 5D) or $\alpha_1 = \alpha_2 = 0.7$ (Fig. S5D).

To clarify the logic leading to $K_m = [S]$, we have deleted the original paragraph below Eq. (10):

Paragraph deleted

To illustrate how Eq. (10) captures the tradeoff relationship between k_2 and K_m , numerical simulations were performed (Fig. 2A). Hereafter, all simulations will be performed at $\alpha_1 = \alpha_{1r} = \alpha_2 = 0.5$, which is a common assumption used to make baseline models in heterogeneous catalysis.^[22,26-28] Physically, this means that when the driving force of an elementary reaction is increased by 1 kJ/mol, its activation barrier decreases by 0.5 kJ/mol. In reality, typical experimental values of α range between 0.3 and 0.7 for artificial catalysts,^[29-31] and the experimental value reported for cellulases is 0.74.^[25] Therefore, the influence of α deviating from 0.5 will be discussed in detail in Fig. 5D.

Instead, the assumption $\alpha_1 = \alpha_2 = 0.5$ now appears much later in the manuscript, between Eqs. (14) and (15). All equations up to Eq. (14) are independent of the α value.

Sentences added (page 8, lines 7-15, blue highlights)

This condition corresponds to a scenario where the activation barriers in the forward and backward directions change equally with respect to the driving force. In general, if the BEP coefficient $\alpha > 0.5$, the forward direction is more sensitive, while if $\alpha < 0.5$, the backward reaction is more sensitive. For reversible enzymes,^[31,32] large deviations from $\alpha = 0.5$ would hinder their ability to catalyze the reaction in both directions. Furthermore, typical experimental values of α range between 0.3 and 0.7 for artificial catalysts,^[33-35] and the experimental value reported for cellulases is 0.74.^[26] Therefore, we expect the unbiased scenario ($\alpha = 0.5$) to be a reasonable representation for the median value of enzymes in general. Setting BEP coefficients to 0.5 is also a common technique used to understand general trends in heterogeneous catalysis.^[22,36-38]

Comment 2-1D

Varying the value of α breaks the optimality relation $K_m = S$.

Reply 2-1D

As noted above, we agree that "maximum activity at $K_m = [S]$ " (Eq. (16)) breaks when $\alpha \neq 0.5$. However, "high activity at $K_m = [S]$ " is correct, regardless of α (Figs. 5 and S5). We hope that the revisions in Reply 2-1C sufficiently clarifies these points.

Comment 2-1E

The bioinformatic analysis they present, if I understand it correctly, is also not especially consistent with $K_m = S$ being a principle seen in nature.

Reply 2-1E

We appreciate this comment by the reviewer, and we have clarified our intent by completely revising the explanation of Fig. 6.

For enzymes which follow the Michaelis-Menten mechanism (red data in Fig. 6, 980 entries), their distribution is centered around $\log_{10} K_m/[S]=0.18$, supporting that $K_m = [S]$ matches what was chosen by natural selection. The standard deviation of $\log_{10} K_m/[S]$ was 1.3, and 53 % of enzymes satisfy $K_m = [S]$ to within an order of magnitude. We expect that this was not clear in the original manuscript, because we had started our explanation from enzymes which deviate $K_m = [S]$ (black and blue data in Fig. 6). Therefore, we have completely rewritten the explanation of Fig. 6 (**page 12, lines 6-37, blue and green highlights**):

Original

This dataset was then classified by the number of entries for each substrate, based on the expectation that a substrate which participates in many reactions is more likely to deviate from Michaelis-Menten kinetics. ATP is the most frequent substrate with 313 entries and is shown in black. Both the raw K_m and S values (Fig. 6A) and their relative distribution (Fig. 6B) shows that $S > K_m$ for ATP. The deviation from $K_m = S$ may be because the Michaelis-Menten mechanism, which is the basis of our mathematical analysis, does not consider scenarios where multiple enzymes must compete for the same substrate, making a smaller K_m more desirable. The next subset shown in blue covers 410 entries and consists of 5 substrates which each appear more than 50 times: NAD^+ , NADH, NADP^+ , NADPH, and acetyl-CoA. These cofactors are less universal than ATP, and S is only slightly larger than K_m . The remaining 980 entries are shown in red. This subset contains 115 substrates such as carbon metabolites and amino acids and appear within the dataset 8 times on average. As the substrate becomes less universal, their K_m and S values become roughly consistent. In particular, the Gaussian distribution fitted to the red histogram (Fig. 6B) has a center at $\log_{10} S/K_m = 0.18$ and a standard deviation of 1.3, which is reasonable considering that influences from inhibitors or the BEP coefficient can change the optimum K_m by roughly an order of magnitude (Fig. 5). Although activity is not the only enzymatic property that must be optimized in nature, the consistency between the K_m of wild-type enzymes and *in-vivo* substrate concentrations suggests that natural selection does indeed favor enzymes which satisfies $K_m = S$, the theoretical guideline for achieving high enzymatic activity.

Revised

This dataset was then classified by the number of entries for each substrate, based on the expectation that a substrate which participates in many reactions is more likely to deviate from Michaelis-Menten kinetics under *in-vivo* conditions. For example, the Michaelis-Menten mechanism does not consider scenarios where multiple enzymes compete for the same substrate, a situation which may occur for cofactors such as ATP. Major metabolites, such as sugars or amino acids all appear less than 50 times each in the dataset and are shown in red. The comparison between their raw K_m and $[S]$ values (Fig. 6A), as well as the histogram of their relative values (Fig. 6B) indicate that the distribution is centered around $K_m = [S]$. Namely, the K_m and $[S]$ are consistent to within 1 order of magnitude for 53% of this dataset (524 out of 980 entries). The Gaussian distribution fitted to the histogram is centered at $\log_{10} K_m/[S] = -0.18$ and has a standard deviation of 1.3. The large standard deviation may be due to a variety of factors, such as inhibitors or BEP coefficients which can change the optimum K_m by roughly an order of magnitude (Fig. 5), growth conditions and measurement errors which may influence $[S]$ also by an order of magnitude,^[43] and the existence of enzymes which do not follow the BEP relationship at all.^[25] Some enzymes operating near the diffusion limit are known to have K_m several orders of magnitude larger than $[S]$. For example, superoxide dismutase has a $K_m > 0.5$ mM,^[44] while the concentration of hydrogen peroxide in aqueous humor is between 25 and 60 μM .^[45] This is a direct contradiction with our proposed law: $K_m = [S]$, which may originate from the fact that the Michaelis-Menten equation is not accurate for enzymes which quickly deplete the substrate (Schnell, FEBS J). However, previous studies^[30] have shown that they are not the majority, and even within our dataset, only 1 % (10 entries) in this dataset show $K_m/[S] > 10^3$. The next subset shown in blue contains 410 entries and consists of 5 substrates which each appear more than 50 times: NAD^+ , NADH, NADP^+ , NADPH, and acetyl-CoA. The Gaussian fitted to the histogram is slightly shifted to smaller K_m (centered at $\log_{10} K_m/[S] = -0.43$), but 57% of this dataset (232 out of 410 entries) still satisfies $K_m = [S]$ to within an order of magnitude. On the other hand, ATP, which is the most frequently occurring substrate with 313 entries, shows a significant deviation from $K_m = [S]$. The fitted Gaussian is centered at $\log_{10} K_m/[S] = -1.64$, and K_m is smaller than $[S]$ for 98% of the entries. The deviation from $K_m = [S]$ may be because the Michaelis-Menten mechanism, which is the basis of our mathematical analysis, does not consider scenarios where multiple enzymes compete for the same substrate. Under such conditions, the effective substrate concentration available to each enzyme would decrease. Thus, $K_m \ll [\text{ATP}]$ may be a result of K_m being adjusted to the effective substrate concentration.

Comment 2-2A

Second, I am concerned about the contribution of this work relative to that of Kari et al. “Physical constraints and functional plasticity of cellulases” *Nat. Commun.* (2021). The present authors cite this work, mostly giving the impression that it is an experimental work, but I was surprised to find that Kari et al. not only provide the optimality principle (in the form $K_m = \frac{\alpha S}{1-\alpha}$), but also a detailed discussion and thermodynamic derivation of where it may come from---an explanation which is substantially the same as (and in my view, better presented) that of the present authors.

Reply 2-2A

We are grateful to this comment by the reviewer, which we have used to clarify our originality. The work by Kari et al found a linear correlation between the k_2 and K_m values of cellulases and identified the existence of a K_m which maximizes activity. However, K_m is a composite parameter which depends on multiple rate constants, and therefore, interpreting K_m in terms of free energies and activation barriers is difficult (see peer review file of Kari et al, referee #2). We speculate this is the reason their group has been using $\Delta\Delta G$ obtained from relative K_m values, instead of raw ΔG values, in this study and also in a previous one (Kari et al., *ACS Catal.*, **2018**, 8, 11966).

Our work attempts to provide the missing link between experimentally accessible parameters (K_m) and the underlying thermodynamics. Namely, we start from the thermodynamic landscape, and then use the BEP and Arrhenius relationships to calculate the activity and K_m . Thus, the concept “high activity at $K_m = [S]$ ” has firm roots in thermodynamics and physical chemistry. We have revised the following sentences to clarify this point (**page 8 lines 22-31, blue highlights**).

Original

Thus, the derivations and simulations so far provide mathematical evidence that having a K_m value equal to the substrate concentration S guarantees maximal enzymatic activity as long as the enzyme follows the Michaelis-Menten mechanism (Scheme 1), and the rate constants follow the BEP relationship with $\alpha_1 = \alpha_{1r} = \alpha_2 = 0.5$.

Revised

Kari et al have reported that the activity of cellulases^[26,39] and PET hydrolases,^[40] are maximized at a specific K_m . However, the physical origin of this trend was unclear, due to the difficulty in obtaining raw ΔG values from experiments. As K_m is a composite parameter depending on multiple rate constants, only relative values of the free energy ($\Delta\Delta G$) have been discussed so far. In this study, we have started from the thermodynamic landscape and have shown that as long as the enzyme follows the Michaelis-Menten mechanism (Scheme 1), and the rate constants follow the BEP relationship with $\alpha_1 = \alpha_2 = 0.5$, tuning the K_m value equal to the substrate concentration $[S]$ guarantees maximal enzymatic activity. The existence of an optimum K_m close to the substrate concentration is valid even under mechanistic deviations as will be shown below.

Comment 2-2B

Importantly, the arguments of Kari et al. seem to me possess at least the same level of generality and rigor as those of the present authors. The present authors assume $\alpha = 0.5$, to get $K_m = S$.

Reply 2-2B

Once again, we appreciate the chance to clarify our originality. The optimality obtained by Kari et al is a specific case of the more general Eq. (13) in this study. Namely, adding an additional assumption ($\alpha_1 = 1 - \alpha_2$) to Eq. (13) yields the optimality by Kari et al ($K_m = \frac{\alpha[S]}{1-\alpha}$). As the reviewer has noted, further assuming $\alpha_1 = \alpha_2 = 0.5$ results in $K_m = [S]$. We have revised the following sentences to clarify these points (**page 11, lines 21-25, blue highlights**).

Original

Although no analytical solution for the optimum K_m was obtained, the optimum K_m under the assumption $\alpha_1 = 1 - \alpha_2$ can be calculated as $\frac{\alpha_2}{1-\alpha_2}S$, which is consistent with the equation reported by Kari et al.

Revised

This dashed line was obtained through numerical optimization, because no analytical solution for the optimum K_m was obtained for general values of α_1 and α_2 . We note that the optimality obtained by Kari et al^[26] ($\frac{\alpha_2}{1-\alpha_2}[S]$) is a special case of Eq. (13), which can be obtained under the assumption $\alpha_1 + \alpha_2 = 1$.

Comment 2-2C

The authors should make much clearer what exactly a reader, especially one familiar with Kari et al., ought to learn from this work. If it is, for example, their numerical simulations loosening the MM assumptions or their bioinformatic analysis, then e.g. this part of the work should be headlined.

Reply 2-2C

We thank the reviewer for the critical comments. The take home message is that if an enzyme operates via the Michaelis-Menten mechanism and satisfies the BEP relationship,

- $K_m = [S]$ yields high activity for general values of α .
- $K_m = [S]$ guarantees maximum activity if $\alpha_1 = \alpha_2 = 0.5$.

We have attempted to clarify both of these points in the revisions noted in Reply 2-1C. In particular, we believe the following new sentences help clarify our intent.

Sentence added (Page 7 line 17, blue highlights)

The fact that $K_m = [S]$ yields high activity is valid even if the BEP coefficients deviate from 0.5 (Fig. 5D, S5).

Sentence added (Page 8 lines 27-31, blue highlights)

... as long as the enzyme follows the Michaelis-Menten mechanism (Scheme 1), and the rate constants follow the BEP relationship with $\alpha_1 = \alpha_2 = 0.5$, tuning the K_m value equal to the substrate concentration $[S]$ guarantees maximal enzymatic activity. The existence of an optimum K_m close to the substrate concentration is valid even under mechanistic deviations as will be shown below.

Sentence added (Page 11, line 21)

The fact that $K_m = [S]$ yields high activity is valid even for other values of α_1 and α_2 (Fig. S5).

Furthermore, to provide the reader a fair assessment on the applicability and limitations of our theory, we have perturbed each of the assumptions:

- Deviations from the Michaelis-Menten equation are provided in Fig. 5A-C and Fig. 6 (ATP)
- Deviations of α values from 0.5 are provided in Fig. 5D and Fig. S5.

We have not performed simulations for when the BEP relationship is not valid at all (no correlation between ΔG and E_a), because the thermodynamic landscape can no longer be converted to rate constants and K_m . However, the existence of such enzymes may be reflected in the large standard deviation of our bioinformatic section, which we have clarified according to Reply 2-1E.

We have also added an explanation on superoxide dismutase as an explicit example where $K_m = [S]$ fails to explain natural selection (Reply 2-1E).

In additions to the revisions concerning the main text noted above, we have revised the abstract to clarify that $K_m = [S]$ is based on basic thermodynamic considerations. We have also toned down our message, because $K_m = [S]$ only yields high (not maximum) activity unless $\alpha_1 = \alpha_2 = 0.5$ (**blue highlights**):

Original (abstract)

Here, we demonstrate that tuning the Michaelis-Menten constant (K_m) to the substrate concentration ($[S]$) maximizes enzymatic activity. This guideline ($K_m = S$) was obtained mathematically by applying the Bronsted (Bell)-Evans-Polanyi (BEP) principle of heterogeneous catalysis to the Michaelis-Menten equation, and is robust even under mechanistic deviations such as reverse reactions and inhibition.

Revised

Here, we demonstrate that tuning the Michaelis-Menten constant (K_m) to the substrate concentration ($[S]$) enhances enzymatic activity. This guideline ($K_m = [S]$) was obtained mathematically by assuming that thermodynamically favorable elementary reactions have higher rate constants, and that the total driving force of the reaction is fixed. Due to the generality of these thermodynamic considerations, we propose $K_m = [S]$ as a general concept to enhance enzymatic activity.

Finally, we have revised the title because the concept “high activity at $K_m = [S]$ ” relies on the BEP relationship which may not be universal.

Original (title)

Universal Design Principle to Enhance Enzymatic Activity using the Substrate Affinity

Revised

Thermodynamic Principle to Enhance Enzymatic Activity using the Substrate Affinity

This concludes our response to reviewer #2. We hope that the revisions so far have helped improve the clarity of our manuscript such that both the advantages and limitations of our theory are clear for the reader.

On behalf of all coauthors, I express my gratitude for the constructive comments by this referee.

Hideshi Ooka (PhD)
hideshi.ooka@riken.jp

REVIEWERS' COMMENTS

Reviewer #1 (Remarks to the Author):

I think that the authors did a great job addressing my comments. In general, I feel that all my comments were dealt with thoroughly. However, one point regarding the validity of the MM-equation remains. This is an important concern since many studies use the MM-eq. outside its validity domain. If the MM-eq. is used outside its validity domain K_m will, in general, be overestimated and trends like the BEP relationship may be overlooked or masked. This study aims to provide a theoretical guideline on how the parameters should be optimized to increase enzymatic activity. As previously, stated I think that this work is both important and relevant but it is equally important to highlight the model's limitations and validity domain if it is to be used by experimentalist.

Reply to "reply 1-2A"

It is expected that the simulations only scale the activities. E_0 is a scaling parameter in the MM-equation so any change in E_0 will just scale the calculated steady-state rate. After all the simulations is based on the MM-eq. and not e.g. a numerical solutions of the ODEs the describe the MM-model.

The reactant stationary assumption is fulfilled when $E_0 \ll S_0 + K_m$. I think that the authors assume that if the substrate does not change due to some preservation mechanism the reaction stationary assumption will automatically be fulfilled. This is not true since substrate depletion can also arise from "trapping" in the enzyme-substrate complex. Substrate preservation ($dS/dt \sim 0$) will ensure steady-state but steady-state does not guarantee that the MM-eq. is valid under a broad range of E_0/S_0 ratios. Even for a simple reversible protein-ligand binding model ($P+L \rightleftharpoons PL$) the solutions $PL = P_0 * L / (L + K_d)$ is only valid for $P_0 \ll L_0 + K_d$. The binding step in both models (MM and protein-ligand) is simplified to a pseudo-first-order reaction by using pseudo-first-order conditions (E.g. $E_0 \ll S_0$ or $P_0 \ll L_0$). To increase the validity domain the authors could solve the MM-model without the pseudo-first-order conditions. This has been done before for systems with "self-preserving" substrates (1) or by using the total quasi-steady-state approximation (2). I would not recommend this since that would narrow the audience of this study. Instead, I think that the authors should state the validity domain of their model so experimentalist do not used it outside this domain. The point is important to me since to many experimental studies is done without paying attention to the validity domain.

It is important to note that I am referring to the use of S_0 (total substrate concentration) in the MM-equation which is custom to do since the free substrate concentration (S) is often not known in an experiment.

References:

1. Andersen, M., Kari, J., Borch, K., and Westh, P. (2018) Michaelis-Menten equation for degradation of insoluble substrate. *Math Biosci* 296, 93-97
2. Tzafiriri, A. R. (2003) Michaelis-Menten kinetics at high enzyme concentrations. *Bulletin of Mathematical Biology* 65, 1111-1129

Reviewer #2 (Remarks to the Author):

I think the revisions made by the authors have significantly improved their manuscript, particularly by clarifying what they are and are not claiming. The takeaway as I understand is that (1) $K_m \sim S$ maximizes activity under some assumptions, and (2) small violations of these assumptions should cause only small violations of optimality conditions, and (3) $K_m \sim S$ might be satisfied within an order of magnitude by many enzymes.

On this last point, I'll just note one thing I noticed, which is that the "non-example" they give, of superoxide dismutase with purportedly $K_m > 0.5 \text{ mM} \gg [S]$ does not seem a straightforward matter to me, as e.g. other works (Bull, C. et al Kinetic studies of superoxide dismutases, *JACS* (1991).) report K_m of tens of micromolar.

Regardless, my original concerns are largely resolved, and I am happy to support publication.

Reviewer #1

Responses to this reviewer are highlighted in yellow and green. Minor English revisions are highlighted in grey. Format changes (section headers, figure numbers, etc) were not highlighted. We have separated and renumbered the comments for clarity.

Overall Comment

General description of the work

I think that the authors did a great job addressing my comments. In general, I feel that all my comments were dealt with thoroughly. However, one point regarding the validity of the MM-equation remains. This is an important concern since many studies use the MM-eq. outside its validity domain. If the MM-eq. is used outside its validity domain K_m will, in general, be overestimated and trends like the BEP relationship may be overlooked or masked. This study aims to provide a theoretical guideline on how the parameters should be optimized to increase enzymatic activity. As previously, stated I think that this work is both important and relevant but it is equally important to highlight the model's limitations and validity domain if it is to be used by experimentalist.

Overall Reply

We are happy to hear that the reviewer found most of our revisions satisfactory. As for the validity domain of the MM-eq, we agree with the reviewer that the model's limitation should be clearly explained. We have revised the manuscript based on the replies below.

Comment 1-1

It is expected that the simulations only scale the activities. E_0 is a scaling parameter in the MM-equation so any change in E_0 will just scale the calculated steady-state rate. After all the simulations is based on the MM-eq. and not e.g. a numerical solutions of the ODEs the describe the MM-model. The reactant stationary assumption is fulfilled when $[E]_0 \ll [S]_0 + K_m$. I think that the authors assume that if the substrate does not change due to some preservation mechanism the reaction stationary assumption will automatically be fulfilled. This is not true since substrate depletion can also arise from "trapping" in the enzyme-substate complex.

Reply 1-1

We appreciate the detailed comments from the referee which shed critical light on the validity and applicability domain of our model. We agree that the MM-eq. is not always valid. Therefore, we have:

- (1) added an explicit sentence saying our model is not applicable when the MM-eq. is invalid.
- (2) provided bovine superoxide dismutase as an example where the MM-eq. breaks down. Hence, this enzyme shows $K_m \neq [S]$.
- (3) provided *T. thermophilus* and *E. coli* as examples of superoxide dismutases which may satisfy the MM-eq. and thus have a K_m close to $[S]$

The revisions were incorporated into the main text are as follows:

Original (page 12 yellow and green highlights):

The large standard deviation may be due to a variety of factors, such as inhibitors or BEP coefficients which can change the optimum K_m by roughly an order of magnitude (Fig. 5), growth conditions and measurement errors which may influence $[S]$ also by an order of magnitude,^[43] and the existence of enzymes which do not follow the BEP relationship at all.^[25] Some enzymes operating near the diffusion limit are known to have K_m several orders of magnitude larger than $[S]$. For example, superoxide dismutase has a $K_m > 0.5$ mM,^[46] while the concentration of hydrogen peroxide in aqueous humor is between 25 and 60 μ M.^[47] This is a direct contradiction with our proposed law: $K_m = [S]$, which may originate from the fact that the Michaelis-Menten equation is not accurate for enzymes which quickly deplete the substrate. However, previous studies^[50] have shown that they are not the majority, and even within our dataset, only 1 % (10 entries) in this dataset show $K_m/[S] > 10^3$.

Revised:

The large standard deviation is due to a variety of factors, such as inhibitors or BEP coefficients which can change the optimum K_m by roughly an order of magnitude (Fig. 6), or growth conditions and measurement errors which may influence $[S]$ also by an order of magnitude.^[43] Furthermore, some enzymes are outside the applicability domain of our model. For example, some enzymes do not follow the BEP relationship at all,^[25] and in some cases, the Michaelis-Menten equation may be an inadequate expression of enzymes under *in-vivo* conditions. Namely, the Michaelis-Menten equation is derived traditionally based on the assumption that the concentration of the enzyme-substrate complex is in the steady state, but this assumption can be broken if the amount or activity of the enzyme is so high such that the substrate is quickly depleted.^[30] Superoxide dismutase from bovine blood is one example, as its high activity ($k_{cat} = 1.9 \times 10^9$ M⁻¹ s⁻¹) renders it to be diffusion-limited^[46] under physiological conditions. Accordingly, it deviates from our proposed law: $K_m = [S]$ with a $K_m (> 0.5$ mM)^[46] several orders of magnitude larger than the substrate concentration ($25 < [H_2O_2] < 60$ μ M in aqueous humor).^[47] Not all superoxide dismutases are exceptions, as those with lower activity ($k_{cat} < 3 \times 10^8$ M⁻¹ s⁻¹) from *Thermus thermophilus* ($K_m=30.8$ μ M)^[48] and *Escherichia coli* ($K_m=75$ μ M)^[49] have K_m values closer to the substrate concentration. Previous studies^[50] have shown that diffusion limited enzymes are not the majority, suggesting that our proposed law may apply to the majority of enzymes. Within our dataset, only 1 % (10 entries) show $K_m/[S] > 10^3$.

Comment 1-2

Substrate preservation ($\frac{dS}{dt} \sim 0$) will ensure steady-state but steady-state does not guarantee that the MM-eq. is valid under a broad range of $[E]_0/[S]_0$ ratios. Even for a simple reversible protein-ligand binding model ($P+L \rightleftharpoons PL$), the solution $[PL] = \frac{[P]_0[L]}{[L]+K_d}$ is only valid for $[P]_0 \ll [L]_0 + K_d$. The binding step in both models (MM and protein-ligand) is simplified to a pseudo-first-order reaction by using pseudo-first-order conditions (E.g. $[E]_0 \ll [S]_0$ or $[P]_0 \ll [L]_0$).

To increase the validity domain the authors could solve the MM-model without the pseudo-first-order conditions. This has been done before for systems with “self-preserving” substrates (1) or by using the total quasi-steady-state approximation (2). I would not recommend this since that would narrow the audience of this study. Instead, I think that the authors should state the validity domain of their model so experimentalist do not used it outside this domain. The point is important to me since too many experimental studies is done without paying attention to the validity domain.

It is important to note that I am referring to the use of $[S]_0$ (total substrate concentration) in the MM-equation which is custom to do since the free substrate concentration ($[S]$) is often not known in an experiment.

References:

1. Andersen, M., Kari, J., Borch, K., and Westh, P. (2018) Michaelis-Menten equation for degradation of insoluble substrate. *Math Biosci* 296, 93-97
2. Tzafiriri, A. R. (2003) Michaelis-Menten kinetics at high enzyme concentrations. *Bulletin of Mathematical Biology* 65, 1111-1129

Reply 1-2

We have read the two papers cited by the reviewer with interest. We agree that directly following their approach, either by using a different mechanism ($E+S \rightleftharpoons ES \rightarrow E+S+P$) or using Riccati equations to solve the differential equations, would diminish the audience of this paper. Therefore, as the reviewer has recommended, we have attempted to clarify the applicability domain of this theory.

During our clarification, we have used the conventional steady state approximation ($d[ES]/dt \approx 0$) as a criterion for the MM-eq. to be valid due to this being the generally accepted criteria today. However, based on Tzafirri (2003, Bull. Math. Biol.) and Schnell (2014, FEBS J.), we acknowledge that the true applicability domain of the MM-eq. may be different. As a detailed discussion on the exact applicability domain of the MM-eq. by distinguishing between standard QSSA (quasi steady-state approximation), reverse QSSA, and total SSA (terminology based on Tzafirri, 2003) seem to be outside the scope of this paper, we have clearly stated that:

- (1) the MM-eq. is a pre-requisite for our theory to be applicable
- (2) the applicability domain is traditionally considered to be the steady state approximation of [ES]
- (3) the exact applicability domain of the MM-eq. is under debate. Citations were added to guide the reader to this discussion.

These revisions were added to the discussion section as follows:

Sentences added (page 14, yellow highlights):

Furthermore, the starting point of our analysis is the Michaelis-Menten equation. Traditionally, this equation has been derived based on the steady state approximation of the enzyme-substrate complex.^[30] Therefore, if this assumption is broken such as in the case of diffusion-limited enzymes,^[46] K_m and $[S]$ may diverge by several orders of magnitude. Recently, several studies have explicitly addressed the differential equations of Michaelis-Menten and similar enzyme mechanisms to determine the exact applicability domain of the Michaelis-Menten equation.^[30,44,45] For example, Schnell^[30] has proposed that instead of the steady-state approximation of [ES], the reactant stationary assumption is the true condition for the Michaelis-Menten equation to be applicable. In this case, the applicability domain of our theory would also adhere to that of the Michaelis-Menten equation.

This concludes our response to reviewer #1. We hope that the revisions so far have helped improve the clarity of our manuscript such that both the advantages and limitations of our theory are clear for the reader.

On behalf of all coauthors, I express my gratitude for the constructive comments by this referee.

Hideshi Ooka (PhD)
hideshi.ooka@riken.jp

Reviewer #2

Responses to this reviewer are highlighted in blue. Minor English revisions were highlighted in grey. Format changes (section headers, figure numbers, etc) were not highlighted.

Overall Comment

I think the revisions made by the authors have significantly improved their manuscript, particularly by clarifying what they are and are not claiming. The takeaway as I understand is that (1) $K_m \sim S$ maximizes activity under some assumptions, and (2) small violations of these assumptions should cause only small violations of optimality conditions, and (3) $K_m \sim S$ might be satisfied within an order of magnitude by many enzymes.

On this last point, I'll just note one thing I noticed, which is that the "non-example" they give, of superoxide dismutase with purportedly $K_m > 0.5 \text{ mM} \gg [S]$ does not seem a straightforward matter to me, as e.g. other works (Bull, C. et al Kinetic studies of superoxide dismutases, JACS (1991).) report K_m of tens of micromolar. Regardless, my original concerns are largely resolved, and I am happy to support publication.

Overall Reply

We are happy to hear that the message of the manuscript is clear now. Points (1)-(3) mentioned by the reviewer are exactly what we would like to propose in this study. Based on the comment on superoxide dismutases (SOD), we have clarified the SOD which we are referring to. Namely, the paper cited in the previous version of our manuscript is a copper-based SOD obtained from bovine blood. In contrast, the paper cited by the reviewer is a manganese-based SOD obtained from *Thermus Thermophilus*. We have revised the manuscript to clarify this point:

Original sentences (green highlights, page 12):

For example, superoxide dismutase has a $K_m > 0.5 \text{ mM}$,^[44] while the concentration of hydrogen peroxide in aqueous humor is between 25 and 60 μM .^[45] This is a direct contradiction with our proposed law: $K_m = [S]$, which may originate from the fact that the Michaelis-Menten equation is not accurate for enzymes which quickly deplete the substrate.^[30]

Revised:

Superoxide dismutase from bovine blood is one example, as its high activity ($k_{cat} = 1.9 \times 10^9 \text{ M}^{-1} \text{ s}^{-1}$) renders it to be diffusion-limited^[46] under physiological conditions. Accordingly, it deviates from our proposed law: $K_m = [S]$ with a K_m ($> 0.5 \text{ mM}$)^[46] several orders of magnitude larger than the substrate concentration ($25 < [\text{H}_2\text{O}_2] < 60 \text{ }\mu\text{M}$ in aqueous humor).^[47] Not all superoxide dismutases are exceptions, as those with lower activity ($k_{cat} < 3 \times 10^8 \text{ M}^{-1} \text{ s}^{-1}$) from *Thermus thermophilus* ($K_m=30.8 \text{ }\mu\text{M}$)^[48] and *Escherichia coli* ($K_m=75 \text{ }\mu\text{M}$)^[49] have K_m values closer to the substrate concentration.

This concludes our response to reviewer #2. On behalf of all coauthors, I express my gratitude for the constructive comments by this referee.

Hideshi Ooka (PhD)
hideshi.ooka@riken.jp